# Immunomodulatory Effect of Rivaroxaban Nanoparticles Alone and in Combination with Sitagliptin on Diabetic Rat Model

**DOI:** 10.3390/diseases13030087

**Published:** 2025-03-19

**Authors:** Mohamed M. Elbadr, Heba A. Galal, Helal F. Hetta, Hassabelrasoul Elfadil, Fawaz E. Alanazi, Shereen Fawzy, Hashim M. Aljohani, Noura H. Abd Ellah, Marwa F. Ali, Ahmed K. Dyab, Esraa A. Ahmed

**Affiliations:** 1Department of Medical Pharmacology, Faculty of Medicine, Assiut University, Assiut 71515, Egypt; mmelbadr@aun.edu.eg (M.M.E.);; 2Department of Pharmacology, Faculty of Veterinary Medicine, Assiut University, Assiut 71515, Egypt; heba.ali@vet.au.edu.eg; 3Division of Microbiology, Immunology and Biotechnology, Department of Natural Products and Alternative Medicine, Faculty of Pharmacy, University of Tabuk, Tabuk 71491, Saudi Arabia; habdelgadir@ut.edu.sa; 4Department of Pharmacology and Toxicology, Faculty of Pharmacy, University of Tabuk, Tabuk 71491, Saudi Arabia; falnazi@ut.edu.sa; 5Department of Medical Microbiology, Faculty of Medicine, University of Tabuk, Tabuk 71491, Saudi Arabia; sh_ibrahim@ut.edu.sa; 6Department of Clinical Laboratory Sciences, College of Applied Medical Sciences, Taibah University, Madina 41477, Saudi Arabia; hsnani@taibahu.edu.sa; 7Department of Pathology and Laboratory Medicine, College of Medicine, University of Cincinnati, Cincinnati, OH 45221, USA; 8Department of Pharmaceutics and Pharmaceutical Technology, Faculty of Pharmacy, Badr University in Assiut, Naser City 2014101, Assiut, Egypt; nora.1512@aun.edu.eg; 9Department of Pharmaceutics, Faculty of Pharmacy, Assiut University, Assiut 71515, Egypt; 10Department of Pathology and Clinical Pathology, Faculty of Veterinary Medicine, Assiut University, Assiut 71515, Egypt; marwa_f_a@aun.edu.eg; 11Department of Medical Parasitology, Faculty of Medicine, Assiut University, Assiut 71515, Egypt; ahmed2015@aun.edu.eg

**Keywords:** diabetes, nano-rivaroxaban, rivaroxaban, sitagliptin, streptozotocin

## Abstract

Background: Chronic inflammation and immune dysregulation are key drivers of diabetes complications. Rivaroxaban (RX) and sitagliptin (SITA) are established therapies for thromboembolism and glycemic control, respectively. This study evaluated the novel therapeutic potential of nano-rivaroxaban (NRX) alone and in combination with sitagliptin (SITA) in mitigating inflammation and restoring immune balance in streptozotocin (STZ)-induced diabetic rats. Methods: Type 2 diabetes was induced in rats using a single injection of STZ (60 mg/kg). Animals were divided into five groups: control, STZ-diabetic, RX-treated (5 mg/kg), NRX-treated (5 mg/kg), and NRX+SITA-treated (5 mg/kg + 10 mg/kg). After 4 weeks of treatment, blood glucose, coagulation markers, pro-inflammatory cytokines (TNF-α, IL-1β, IL-6), and anti-inflammatory cytokines (IL-35, TGF-β1, IL-10) were analyzed. Histopathological examination of the liver, kidney, pancreas, and spleen was conducted. Immunohistochemistry was used to assess hepatic NF-κB expression. Results: STZ significantly elevated pro-inflammatory cytokines (IL-1β, TNF-α, IL-6) and anti-inflammatory cytokines (IL-35, TGF-β1, IL-10), along with increased hepatic NF-κB expression and histopathological abnormalities in immune organs. NRX significantly reduced inflammatory cytokines, improved histopathological changes in organs, and decreased hepatic NF-κB expression. The combination therapy (NRX + SITA) achieved superior immune modulation, with enhanced cytokine profile restoration, reduced hepatic NF-κB expression, and near-complete histopathological normalization. Conclusions: This study underscores the promise of combining nanoparticle-based drug delivery with established therapies like sitagliptin to achieve superior immune modulation and inflammation control, presenting a potential therapeutic strategy for managing diabetes complications.

## 1. Introduction

Type 2 diabetes (T2D) is a major global health concern, contributing to complications such as cardiovascular disease, nephropathy, and neuropathy. It is characterized by elevated blood glucose levels due to insulin resistance or insufficient insulin production. Long-term, uncontrolled diabetes can lead to serious complications, such as damage to the kidneys, heart, blood vessels, and eyes [1]. Among these, thrombosis is a significant concern for diabetic patients, as increased glucose levels promote vasculitis and thrombus formation [2]. Chronic inflammation plays a key role in the progression of T2D, with dysregulated immune responses leading to insulin resistance and organ damage. A complex network of pro-inflammatory and anti-inflammatory cytokines regulates this inflammation. Thus, reducing inflammation and restoring immune function are critical strategies for managing and preventing these complications [3].

Rivaroxaban (RX), a direct factor Xa inhibitor, is an anticoagulant that reduces thrombus formation by inhibiting platelet and leukocyte adhesion to the endothelial wall. Beyond its anticoagulant properties, RX also exerts anti-inflammatory effects, improving atherosclerotic lesions in experimental models [4,5,6].

Nanotechnology offers a promising approach for targeted drug delivery to specific cells or tissues in the body. Nanoparticles improve pharmacokinetics, enhance drug efficacy, and minimize adverse effects by reducing the frequency of administration and enabling precise interaction with cellular structures [7]. Their small size allows for efficient penetration and accumulation in target cells, facilitating modifications in cellular processes that can aid in managing diseases like diabetes, cancer, and renal disorders [8]. In this regard, rivaroxaban nanoparticles (NRX) hold significant potential for advancing therapies by enhancing tissue targeting, improving drug delivery, and reducing side effects.

Diabetic patients often require oral anticoagulants to prevent or manage thromboembolic complications; however, some anticoagulants, such as warfarin, are associated with an increased risk of hypoglycemia. Therefore, selecting an oral anticoagulant with a lower hypoglycemia risk is crucial. Non-vitamin K antagonist oral anticoagulants (NOACs), including rivaroxaban, edoxaban, dabigatran, and apixaban, have demonstrated a significantly lower risk of severe hypoglycemia compared to warfarin, particularly in patients receiving antidiabetic treatments such as sulfonylureas, metformin, or dipeptidyl peptidase-4 inhibitors. Among these, rivaroxaban has shown promising anticoagulant and anti-inflammatory properties, making it a potential therapeutic option for diabetic patients requiring anticoagulation therapy [9,10].

Sitagliptin (SITA), a dipeptidyl peptidase-4 (DPP-4) inhibitor, is a widely used drug for managing Type 2 diabetes by increasing the levels of incretin hormones, which promote insulin secretion [11]. Beyond its glucose-lowering effects, SITA has shown antioxidant, anti-inflammatory, and anti-apoptotic properties [12]. Subsequently, combining NRX with SITA could offer synergistic effects in managing diabetic complications by targeting inflammation and restoring immune functions.

This study aims to evaluate the immunomodulatory effects of RX, its nanoparticle formulation (NRX), and the combination of NRX and SITA in an STZ-induced diabetic rat model. We assessed the impact on pro-inflammatory (TNF-α, IL-1β, IL-6) and anti-inflammatory cytokines (TGF-β, IL-10, IL-35), behavioral changes, coagulation profiles, histopathological alterations in key organs, and the expression of NF-κB in the liver through immunohistochemical analysis.

## 2. Materials and Methods

### 2.1. Materials and Chemicals

#### 2.1.1. Drugs and Chemicals

Rivaroxaban powder (Sigma-Aldrich, St. Louis, MO, USA) was dissolved in 0.5% carboxymethyl cellulose (CMC), while nicotinamide powder (Sigma-Aldrich, St. Louis, MO, USA) was prepared as a 4% solution in saline. Streptozotocin (STZ) powder (Sigma-Aldrich, St. Louis, MO, USA) was dissolved in 0.1 mol/L cold citrate buffer (pH 4.5), and sitagliptin powder (Merck Sharp and Dohme Ltd., Pavia, Italy) was dissolved in saline. Sodium citrate, citric acid, and sodium carboxymethyl cellulose (Na-CMC) were obtained from ADWIC Co. (Cairo, Egypt).

#### 2.1.2. ELISA Kits

ELISA kits for rat TNF-α, IL-6, IL-1β, IL-10, and TGF-β were sourced from DLDEVELOP (China), while the IL-35 kit was obtained from SUNRED (Wuxi, China).

#### 2.1.3. Nanoparticle Materials

For nanoparticle preparation, PLGA (25,000–35,000 Da, lactic acid–glycolic acid = 60:40) was purchased from Polyscitech-Akina Inc. (Lafayette, IN, USA), and Pluronic F-127 (PF-127) was sourced from Sigma Chem. Co. (St. Louis, MO, USA).

### 2.2. Ethical Approval

The study protocol followed globally recognized guidelines outlined in the “Guide for the Care and Use of Laboratory Animals”. The Assiut Veterinary Ethics Committee approved the study (Approval No. 06/2024/0153).

### 2.3. Animals and Experimental Groups

#### 2.3.1. Animals and Housing Conditions

Fifty male Wistar rats (180–220 g) were obtained from the Assiut University Faculty of Veterinary Medicine’s animal house. The rats were housed in stainless steel cages at 25 °C under a 12 h light/dark cycle with unlimited access to food and water. The inclusion criteria for the animals required male Wistar rats weighing more than 180 g and having a random blood glucose level exceeding 250 mg/dL three days after STZ injection. Rats were excluded if they weighed less than 180 g or had a random blood glucose level below 250 mg/dL after STZ administration.

#### 2.3.2. Animal Groups

The study included five groups of male Wistar rats (10 animals per group). The control non-diabetic group received 0.5% carboxymethyl cellulose (CMC; 2 mL/kg) via stomach tube for four weeks [13]. In the second group, Type 2 diabetes was induced using a single intraperitoneal injection of streptozotocin (STZ; 60 mg/kg) following pretreatment with nicotinamide (110 mg/kg, i.p.), and these rats remained untreated throughout the study [14,15]. The third group, STZ + RX-treated, received oral administration of rivaroxaban (RX; 5 mg/kg) for four weeks, while the fourth group, STZ + NRX-treated, was given nano-rivaroxaban (NRX; 5 mg/kg RX equivalent) orally for the same duration [13]. The fifth group received a combination of NRX (5 mg/kg) and sitagliptin (SITA, 10 mg/kg) for four weeks [16,17].

### 2.4. Methods

#### 2.4.1. Fabrication and Characterization of RX/PLGA Nanoparticles

Nanoparticles were synthesized as previously described by Hetta et al. [18]. Briefly, RX and PLGA were dissolved in acetone (1 mL) and mixed at different weight ratios (1:1, 1:3, 1:5). The solution was dropped into a 1% *w/v* aqueous Pluronic (PF-127) solution under magnetic stirring and left overnight. Nanoparticles were collected via centrifugation at 14,000 rpm (4 °C) for 20 min.

After collecting the nanoparticles, the entrapment efficiency (EE%) of the encapsulated RX was measured indirectly by spectrophotometric analysis of the supernatant at 255 nm. Then, the particle size and Polydispersity Index (PDI) were determined using a Zetasizer Nano-ZS (Malvern Instruments, Malvern, UK).

#### 2.4.2. Induction of Type 2 Diabetes Mellitus

Type 2 diabetes was induced in rats by an i.p. injection of 110 mg/kg nicotinamide, followed 15 min later by a single i.p. injection of 60 mg/kg STZ prepared in 0.1 mol/L cold citrate buffer (pH 4.5) [14,15]. To prevent hypoglycemia-induced mortality, the animals were given a 5% glucose solution instead of water for 24 h post-STZ injection. Three days after the STZ injection, blood glucose levels were measured using a hand-held glucometer (BIONIME CORPORATION, Dali City, Taiwan). Rats with random blood glucose levels greater than 250 mg/dL were considered diabetic and used for the study [15,19].

#### 2.4.3. Evaluation of the Behavioral Activity

A rotarod apparatus (Ugo Basile, Varese, Italy) was used for the determination of motor coordination. Rats were placed on the rotating rod apparatus twice; once on day 0 and the other at the end of week 4 of the experiment. The time of staying on the rod was recorded for all rats. Rats that stayed for 60 s (cut-off time) on the rotarod bar were included in the study. The mean of 2 trials was used for statistical analysis [20,21,22,23].

The thermal pain threshold was evaluated using the hot plate test. Animals were placed on a hot plate maintained at 55 °C, and the reaction time from the heat stimulus to the response (jumping or licking of the hind paws) was recorded. To prevent tissue damage, a 15 s cut-off time was applied. The hot plate test was performed at both the beginning and the end of the study to assess the effect of the tested drugs [22,23,24].

#### 2.4.4. Coagulation Profile Tests

Prothrombin time (PT) was measured using Diagen calcium brain thromboplastin reagent [25]. Activated partial thromboplastin time (aPTT) was determined using a Diagen kaolin platelet substitute mixture [26,27]. These tests were operated according to the manufacturer’s instructions. To assess bleeding time, the rat’s tail was first warmed for 1 min in 40 °C water, then a small incision was made. The filter paper was used to blot the incision every 15 s until the bleeding stopped, with the time at which bleeding ceased recorded as the bleeding time [27,28,29].

#### 2.4.5. Animal Sacrifice and Sample Preparation

At the end of the study, animals were anesthetized with 4% isoflurane [30]. Blood samples were obtained through cardiac puncture, and the animals were then sacrificed by cervical decapitation. The collected blood was centrifuged at 3000 rpm for 10 min to separate the serum [31], which was subsequently stored at −80 °C for the analysis of inflammatory markers [32]. Additionally, internal organs such as the kidney, liver, pancreas, and spleen were harvested for histopathological and immunohistochemical examination.

#### 2.4.6. Evaluation of Pro-Inflammatory and Anti-Inflammatory Cytokines in Serum

The concentrations of pro-inflammatory cytokines (IL-1β, IL-6) and anti-inflammatory cytokines (IL-10, TGF-β, IL-35) in rat serum samples were quantified using commercially available enzyme-linked immunosorbent assay (ELISA) kits. The kits used included IL-1β (Cat. No. DL-TNFa-Ra, DLDEVELOP, Wuxi, China), IL-6 (Cat. No. DL-IL6-Ra, DLDEVELOP, China), IL-1β (Cat. No. DL-1L1b-Ra, DLDEVELOP, China), IL-10 (Cat. No. DL-IL10-Ra, DLDEVELOP, China), TGF-β (Cat. No. DL-TGFB1-Ra, DLDEVELOP, China), and IL-35 (Cat. No. SRB-T-88705, SUNRED, China). The assays were performed according to the manufacturer’s protocols, and optical density (OD) was measured at a wavelength of 450 nm using a SPECTROstar Nano microplate reader (BMG Labtech, Ortenberg, Germany). Each serum sample was analyzed in triplicate, and the results, expressed as cytokine concentrations in pg/mL, represent the mean of these measurements.

#### 2.4.7. Histopathological Examination

Tissue samples from the kidney, liver, pancreas, and spleen were fixed, dehydrated in ascending grades of ethyl alcohol, and cleared with xylene before being embedded in paraffin wax. Paraffin blocks were sectioned into 4 μm slices, mounted on slides, and dried overnight at 37 °C before hematoxylin and eosin (H&E) staining. The stained sections were examined under a light microscope (Olympus CX31; Tokyo, Japan), and images were captured using a digital camera (Toupview, LCMos10000KPA, China) [33].

Liver damage was categorized into vascular findings and necrobiotic changes, with lesions scored as absent (0), slight (1), moderate (2), or severe (3) using 40× magnification [33]. Renal injury was evaluated for glomerular, tubular, and interstitial changes at 40× magnification [34]. Pancreatic injury was assessed based on cell vacuolization, acinar cell necrosis, and inflammatory cell infiltration, also using 40× magnification [35].

#### 2.4.8. Immunohistochemical Evaluation of Hepatic NF-κB

Paraffin-embedded liver sections on charged slides were used for the immunohistochemical detection of hepatic NF-κB. The sections were first deparaffinized, rehydrated, and subjected to antigen unmasking using a citrate buffer in a water bath. Endogenous peroxidase activity was then blocked with 3% H_2_O_2_. Then, the sections were incubated with a diluted polyclonal primary antibody for NF-κB (Cat. No. E-AB-13815, Elabscience, Houston, TX, USA) for 30 min, followed by incubation with a secondary antibody (Cat. No. AMF-080-IFU, ScyTek, North Hollywood, CA, USA) and treatment with diaminobenzidine (DAB) chromogen. Finally, they were stained with hematoxylin, dehydrated, and mounted.

Immunopositive cells and the total number of hepatocytes were counted in five fields of each hepatic section using an Axiostar Plus microscope (Carl Zeiss, Thornwood, NY, USA) equipped with an Axiostar Plus digital camera and Axiovision 4.1 software [36].

### 2.5. Statistical Analysis

Data were expressed as mean ± standard error of the mean (SEM). One-way analysis of variance (ANOVA) followed by Tukey’s post hoc test was used to evaluate differences between groups. Significance was set at *p* < 0.05. All statistical analyses were performed using GraphPad Prism 9 software (GraphPad, San Diego, CA, USA).

## 3. Results

### 3.1. Fabrication and Characterization of RX/PLGA Nanoparticles

RX was encapsulated inside PLGA polymer using the nano-precipitation method [18,37,38]. Nanoprecipitation is a simple and reproducible method that gives particles of different sizes. PLGA (poly (lactic-co-glycolic acid)) is a synthetic, biocompatible, and biodegradable polymer commonly used in nanoparticle formulations. The drug and polymer were dissolved in acetone at different weight ratios to test PLGA’s encapsulation ability. The organic phase was then injected into an aqueous phase containing PF-127 to prevent nanoparticle aggregation. The addition of the organic phase caused the transparent solution to turn milky white, indicating nanoprecipitation. All RX/PLGA nanoparticles exhibited high entrapment efficiency, exceeding 90%. Particle size increased with the PLGA content. The RX/PLGA nanoparticles prepared at a 1:3 weight ratio, with an entrapment efficiency of 97.03%, a size of 379.15 nm, and a PDI of 0.43, were selected for further in vivo evaluation (Table 1).

### 3.2. Evaluation of Blood Glucose Level

The results showed that i.p. injection of STZ significantly increased blood glucose levels from the third day of injection until the end of the study (4 weeks) in all animal groups, compared to the control group. The combination of NRX + SITA led to a significant decrease in blood glucose levels after one, two, three, and four weeks of STZ injection, compared to the groups receiving either STZ or NRX alone (Figure 1).

### 3.3. Evaluation of Behavioral Activities

A single i.p. injection of streptozotocin (60 mg/kg) significantly reduced rotarod staying time in all animal groups compared to the control non-diabetic group. However, combined treatment with NRX + SITA for 4 weeks significantly improved the rotarod staying time in diabetic rats, as compared to the STZ and NRX-only treated groups. No significant difference was observed in the rotarod performance between the RX and NRX groups relative to the STZ diabetic group (Figure 2).

In terms of pain sensitivity, streptozotocin-induced diabetes resulted in hyperalgesia, evidenced by a significant reduction in hot plate latency after 4 weeks of treatment. This reduction was observed in the STZ, RX, and NRX groups compared to the control group. However, combined NRX + SITA treatment significantly enhanced hot plate latency in diabetic rats when compared to the STZ- and NRX-only groups. No significant difference in hot plate latency was found between the RX and NRX-treated groups compared to the STZ diabetic group (Figure 3).

### 3.4. Coagulation Profile Tests

The prothrombin time (PT) was significantly reduced in the STZ-induced diabetic group compared to the control non-diabetic group. In contrast, treatment with RX, NRX, and the combination of NRX + SITA led to a significant prolongation of PT, demonstrating an improvement in coagulation compared to the diabetic rats (Figure 4A).

Similarly, STZ administration resulted in a notable decrease in aPTT when compared to control non-diabetic rats. All the tested drugs—RX, NRX, and NRX + SITA—induced a significant increase in APTT relative to the STZ diabetic group, with NRX showing a more pronounced effect than RX (Figure 4B).

Regarding BT, a significant reduction was observed in the STZ diabetic group compared to non-diabetic rats. However, treatment with RX, NRX, and NRX + SITA resulted in a substantial increase in BT, significantly improving clotting function compared to the diabetic group (Figure 4C).

### 3.5. Evaluation of Pro-Inflammatory and Anti-Inflammatory Cytokines in Serum

This study evaluated the impact of RX, NRX, and their combination with SITA on serum levels of pro-inflammatory (IL-1β, IL-6, TNF-α) and anti-inflammatory cytokines (TGF-β, IL-10, IL-35) in STZ-induced diabetic rats. STZ injection significantly increased all measured cytokines compared to control rats, indicating an inflammatory response.

Treatment with NRX (5 mg/kg orally) and the combination of NRX (5 mg/kg) with SITA (10 mg/kg) for four weeks resulted in a notable reduction in cytokine levels, with the combination therapy consistently showing the most pronounced effects. Specifically, IL-1β and IL-6 levels, elevated after STZ injection, were significantly decreased following NRX treatment, with the greatest decline observed in the combination group (Figure 5A,B). Similarly, TNF-α levels showed substantial reductions across all treatment groups, with the combination therapy yielding the most significant decrease (Figure 5C).

For anti-inflammatory cytokines, TGF-β, IL-10, and IL-35 levels, which were elevated post-STZ injection, also declined markedly after treatment. The combination therapy consistently produced the greatest reduction in these cytokines compared to individual treatments (Figure 5D–F).

### 3.6. Histopathological Examination

#### 3.6.1. Effect of the Tested Drugs on Histopathology of the Liver

Histopathological examination of the liver in the control group revealed normal hepatic architecture (Figure 6A). In contrast, the STZ-induced diabetic group exhibited significant pathological changes, including fat degeneration characterized by clear fat vacuoles in hepatocytes, blood vessel congestion, and peri-portal infiltration with inflammatory cells, accompanied by Kupffer cell proliferation (Figure 6B–D).

In the RX-treated group, these changes were somewhat alleviated, with mild congestion of the central vein and peri-portal lymphocytic infiltration observed (Figure 6E,F). The NRX-treated group showed improvement, with most hepatic alterations reduced, though some instances of vacuolar degeneration remained (Figure 6G). The combination treatment of NRX + SITA demonstrated near-normal hepatic architecture, with normal hepatic cords observed, indicating substantial recovery (Figure 6H).

#### 3.6.2. Effect of the Tested Drugs on Kidney Histopathology

Histopathological examination of the kidney in the control group revealed normal glomeruli and renal tubules (Figure 7A). In contrast, the STZ-induced diabetic group exhibited significant pathological changes, including the presence of clear fat vacuoles in the renal tubular epithelium, the complete absence of glomerular space, and the appearance of hyaline casts in the lumen of the renal tubules. Additionally, vascular changes were noted, characterized by interstitial hyperemia (Figure 7B–D).

In the RX-treated group, epithelial casts were detected in most cases (Figure 7E). The NRX-treated group showed mild abnormalities, such as the absence of glomerular space, while most renal tubules appeared normal (Figure 7F,G). Notably, the NRX + SITA-treated group exhibited a normal glomerular structure, indicating a significant improvement (Figure 7H).

#### 3.6.3. Effect of the Tested Drugs on Pancreas Histopathology

Histopathological examination of the pancreas in untreated animals revealed normal Langerhans islets with pale, rounded, and ovoid β-cells at the center (Figure 8A). In the STZ-induced diabetic group, significant pathological changes were observed, including atrophy and vacuolar degeneration of the Langerhans islet cells, accompanied by inflammatory cellular infiltration around the peri-lobular duct (Figure 8B–D).

The RX-treated group showed mild vacuolar degeneration in the Langerhans islet cells, along with congestion of interstitial blood vessels (Figure 8E,F). In the NRX-treated group, atrophy of the Langerhans islets was evident (Figure 8G). However, the NRX + SITA-treated group demonstrated normal islets of Langerhans, indicating a significant improvement in pancreatic histopathology (Figure 8H).

#### 3.6.4. Effect of the Tested Drugs on Spleen Histopathology

Microscopic examination of the spleen in the control group revealed a normal structure, including a white pulp with a central arteriole and a red pulp (Figure 9A). In contrast, the STZ-induced diabetic group exhibited severe hemosiderosis in the spleen (Figure 9B,C).

No significant abnormalities were observed in the spleen of the RX-treated, NRX-treated, or NRX + SITA-treated groups, with all groups showing normal spleen architecture (Figure 9D–F).

#### 3.6.5. Histopathological Scoring Results

Histopathological scoring revealed significant improvements in liver, kidney, and pancreatic tissues across the treatment groups, particularly in the NRX + SITA combination group.

In the liver, necrobiotic changes such as vacuolar degeneration, fatty degeneration, and necrosis were observed alongside vascular abnormalities, including sinusoidal dilatation, endothelial desquamation, and hemorrhages. These pathological features were significantly reduced in the RX, NRX, and NRX + SITA groups compared to the control, with the NRX + SITA combination group showing the most marked improvement, as illustrated in Figure 10A.

Similarly, kidney histopathology showed glomerular damage (e.g., congestion, periglomerular proliferation, thickened Bowman’s capsule, and narrowed Bowman’s space), tubular injury (e.g., epithelial vacuolation, dilatation, casts, and necrosis), and interstitial changes (e.g., inflammation, congestion, and hemorrhages). These abnormalities were significantly alleviated in all treatment groups, with the NRX + SITA combination group demonstrating the greatest reduction in renal pathological changes, as shown in Figure 10B.

Findings in the pancreas included acinar cell necrosis, vacuolization, and inflammatory cell infiltration in the intralobular areas. Treatment with RX, NRX, and NRX + SITA significantly reduced these pathological alterations, with the NRX + SITA combination group showing the most substantial improvement, as depicted in Figure 10C.

### 3.7. Immunohistochemical Evaluation of Hepatic NF-κB

The immunohistochemical analysis of NF-κB, an indicator of inflammatory response in hepatic tissue, revealed a negative reaction in the control group, indicating the absence of inflammation (Figure 11A). In contrast, the STZ-induced diabetic group exhibited severe NF-κB expression across most fields, consistent with heightened inflammatory activity (Figure 11B,C).

Moderate NF-κB expression was detected in the RX- and NRX-treated groups, suggesting a partial reduction in inflammation compared to the diabetic group (Figure 11D,E). Notably, the NRX + SITA-treated group demonstrated only mild NF-κB expression in the majority of hepatic tissue sections, reflecting significant amelioration of inflammatory responses (Figure 11F).

## 4. Discussion

Given the documented immunomodulatory and anti-inflammatory properties of RX, this study aimed to assess the effects of RX and NRX on the immune system in diabetic rats. This was achieved by measuring the concentrations of key cytokines, including IL-6, IL-1β, IL-10, IL-35, TNF-α, and TGF-β. Furthermore, the investigation extended to evaluating histopathological alterations in immune-related organs such as the spleen, liver, kidney, and pancreas.

The study indicated that an STZ injection of 60 mg/kg resulted in hyperglycemia and hyperalgesia in diabetic animals determined by the hot plate test. The results were in harmony with the report of Forman et al. and Sharma et al. who showed that STZ 65 mg induced hyperglycemia and reduced hot plate latency after 4 weeks of injection [23,39]. Streptozotocin is a commonly used agent to induce diabetes in experimental animals. Destruction of the β-cells of the pancreas, excess production of glucose, and decreased glucose utilization are the possible causes of hyperglycemia induced by streptozotocin [40]. Although the precise mechanism by which hyperglycemia reduces hot plate latency and causes hyperalgesia is unknown, many theories have been given, as hyperinsulinemia induced by hyperglycemia may be responsible for the reduction in pain threshold [41].

In this study, the administration of nano-rivaroxaban and sitagliptin for 4 weeks in diabetic rats caused a remarkable rise in hot plate latency and rotarod staying time. These results were in line with Sharma et al. who observed that the diabetic animals treated with sitagliptin 11.67 mg/kg showed an improvement in motor behavior and pain sensitivity [23]. 

The study results showed that PT, aPTT, and BT decreased remarkably after STZ injection. The results were in line with ElGendy and Abbas who reported that 70 mg/kg STZ significantly decreased PT and aPTT [26], and Ayodele et al. who reported that streptozotocin (55 mg/kg) decreased PT, aPTT, and BT [27]. Keskin and Uluışık reported similar results as they indicated that S.C STZ 40 mg/kg for two days reduced PT and aPTT [42]. 

Keskin and Uluışık suggested that hyperglycemia caused changes in thrombocytes, dysfunction of the endothelium, and enhanced platelet aggregation, which induced a procoagulant state in diabetes [42]. According to Yeom et al., hyperglycemia elevates levels of fragments of prothrombin, with a reduction in factor VII and elevation in factor VIII [43].

In this study, the administration of NRX in diabetic animals demonstrated an increase in PT, aPTT, and BT. Liu et al. reported similar results as they indicated that rivaroxaban caused an increase in prothrombin time [44] Also, Tinel et al. agreed with these results, as they indicated that rivaroxaban 2 mg/kg in rats increased PT and BT [45] Since factor X inhibitor action is mediated in the two coagulation pathways, it is expected that rivaroxaban can affect both PT and aPTT, so it prolongs both PT and aPTT in a dose-dependent manner. However, PT seems to be more sensitive to rivaroxaban [46].

Atherosclerosis is associated with injury in the endothelium, inflammatory changes, accumulation of lipids, and formation of plaque inside the vessel wall intima. Atherosclerotic plaque rupture is the main reason for acute atherothrombotic accidents with the resulting occlusion of vessels [47]. Coagulation of blood and atherosclerosis are associated with each other. Some proteins involved in coagulation, such as factor VII and tissue factor, are included within the atherosclerotic plaque. These factors are involved in the pathogenesis of atherosclerotic processes such as angiogenesis and inflammation [48]. Currently, the management of diabetic-induced atherosclerosis focuses on developing new anticoagulant drugs. Rivaroxaban is approved for the treatment of atherosclerosis induced by diabetes and decreases its complications. The principle behind this suggestion is the inhibition of factor Xa [49].

Cytokines are proteins released by various cell types that regulate and mediate immune system responses. They play a crucial role in the pathogenesis of diabetes [50]. Research has shown that insulin resistance is closely associated with the excessive production of pro-inflammatory cytokines, including IL-6, IL-1β, and TNF-α, which contribute to disease progression. In contrast, anti-inflammatory cytokines have been suggested to play a compensatory role in modulating immune responses in diabetes [50,51].

The results of this study indicated that STZ injection resulted in a remarkable elevation of different cytokines. Jain et al. agreed with the current results as they demonstrated that STZ 65 mg/kg in rats caused a significant rise in pro-inflammatory cytokines, IL-6, TNF-α, and IL-1β [52]. Moreover, Sindhughosa and Pranamartha reported also that serum pro-inflammatory cytokines were elevated in the STZ-induced diabetic rats, indicating an increase in the activity of inflammation [53]. 

In this study, the use of RX, which is an inhibitor of factor Xa, caused a decline in the level of different cytokines. Rivaroxaban may improve atherosclerosis by reducing TNF-α and IL-6 and interfering with macrophage activation [49]. In addition, factor Xa increases pro-inflammatory cytokines secretion such as IL-8, IL-6, and MCP-1 [21].

When diabetic rats were treated with NRX or a combination of NRX + SITA for 4 weeks, serum concentration of pro-inflammatory cytokines (TNF-α, IL-6, and IL-1β) significantly declined, which is in line with the results indicated by Zhou et al. who reported that administration of rivaroxaban in a dose of 5 mg reduced mRNA expression of TNF-*α* and IL-6 [5]. In addition, these results were similar to those reported by Ferreira et al. who indicated that sitagliptin reduced TNF-*α* and IL-1β in rats with diabetes [54].

The findings of this study indicated that STZ caused a remarkable elevation of IL-10, which is in harmony with the report of Gouda et al. who indicated that the rise in IL-10 in diabetics is a compensatory mechanism for the increase in pro-inflammatory cytokines [55]. IL-10 has a significant role in immune system regulation as it decreases the production of cytokine, suppressing the expression of tissue factor and enhancing the shifting of lymphocytes to the T-helper 2 phenotype [56].

The results showed that STZ caused a noticeable rise in IL-35. Recent research has suggested that IL-35’s protective effects are attributable to reducing hyperglycemia and blocking the attack of inflammatory factors. IL-35 anti-inflammatory effects are caused by macrophage phenotype regulation, inhibition of proliferation of T cells, blocking of T-helper 17 cell differentiation, and enhancing IL-35-producing regulatory T cells (iTr35) [57].

TGF-β1 was elevated after STZ injection in this study in coordination with Hussain et al. who documented that TGF-β1 was increased in cases of peripheral diabetic neuropathy [58]. TGF-β1 is an anti-inflammatory cytokine that blocks macrophage activation by interfering with Toll-like receptor-dependent pathways signaling [58]. Pro- and anti-inflammatory cytokines are involved during the activation of the immune system in Type 2 diabetes development. The elevated anti-inflammatory cytokines may act as a compensatory mechanism against the pro-inflammatory cytokines [55].

Rivaroxaban blocks the production of reactive oxygen species and reduces the expression of the MCP-1 gene in late glycation end products via antagonizing the thrombin/PAR-2 system. Rivaroxaban lessens the progression of atherosclerotic plaques and destabilization in ApoE (-/-) mice by blocking the activation of pro-inflammatory cytokines released from macrophages. This study indicated that rivaroxaban may be useful in the treatment of both atherosclerosis and thrombosis [59].

Nanomedicine, which is the incorporation of nanotechnology with medicine, has been widely employed to give an entirely new outlook to current medications. Nanoparticles are a visually appealing method for better controlling hyperglycemia because of their benefits, which include prolonged drug release, a reduction in the frequency of administration, a decline in adverse effects, and the ability to facilitate cellular/molecular interactions because of their small size [7].

In this study, diabetic animals treated with NRX for 4 weeks demonstrated a significant decline in pro-inflammatory (IL-1, TNF-α, and IL-6) and anti-inflammatory cytokines (IL-35, IL-10, and TGF-β1) when compared with the untreated diabetic or rivaroxaban-treated group. Adding sitagliptin to NRX in this study caused a greater decline in IL-6, IL-1, and TNF-α in comparison to a group of rats that received NRX alone. These results were in line with Ferreira et al. who indicated that sitagliptin use decreased inflammatory cytokines in rats [54]. Also, Tremblay et al. were in agreement with these results as they reported that sitagliptin use for 6 weeks reduced IL-1 and TNF-α [60]. Sitagliptin monotherapy significantly decreased triacylglycerol, and free fatty acid levels and elevated HDL-cholesterol in diabetic patients [61]. Furthermore, sitagliptin reduces blood pressure in nondiabetic hypertensive patients [62], so sitagliptin is anti-inflammatory and lessens the progress of atherosclerosis [63]. Dobrian et al. in line with these results, reported that in a diet obesity model rat, treatment with sitagliptin reduced the expression of adipocyte mRNA of IL-6, TNF-α, and IL-12, indicating that sitagliptin may have a remarkable anti-inflammatory effect [64].

Type 2 diabetes and insulin resistance are associated with endothelial dysfunction and inflammatory processes. More precisely, cardiovascular disease and T2D have been linked to increased levels of circulating cytokines, especially interleukins and C-reactive protein [65]. The beneficial role of sitagliptin on endothelial dysfunction and inflammatory processes may be attributable to the elevation of the level of Glucagon-like peptide-1 (GLP-1) that inhibits inflammation in blood vessels. The protective role of GLP-1 in cases of endothelial dysfunction induced by TNF-α is mediated by adjustment of plasminogen activator inhibitor-1 mRNA, adhesion molecules expression, and protein secretion [66]. Incretin pathway activation stimulates the proliferation of neuronal cells and inhibits the death of cells. Sitagliptin inhibits GLP-1 metabolism, which leads to protective effects on neurons in diabetic animals. It also has anti-inflammatory, antioxidant, and anticancer effects [25,67].

However, sitagliptin’s anti-inflammatory activity not only depends on GLP-1 but also regulates the expression of NF-κB. It inhibits the pathway of TNF-α/ICAM-1/VCAM-1 with cell function protection independent of GLP-1 [68]. Sitagliptin may possess anti-inflammation activity by regulating the signaling pathway of MAPK and AMPK [69]. Sitagliptin was reported to improve endothelial and monocyte function and decrease atherosclerosis lesions in mice with a deficiency in apolipoprotein E [63]. Dipeptidyl peptidase-IV is responsible for the metabolism of several substances, such as different chemokines that participate in the response to innate immunity. Inhibition of this enzyme leads to a direct inhibitory effect on monocyte migration and CD4-positive T cells that are involved in atherogenesis lesions [63].

Histopathological examination of the liver in this study demonstrated that STZ caused degeneration in hepatocytes, congestion of blood vessels, and peri-portal infiltration with inflammatory cells accompanied by Kupffur cell proliferation. Proliferation and inflammation of Kupffer cells are attributable to the liver’s role in the regulation of glucose homeostasis via modification of hepatokines expression. Fetuin, which is one of the important hepatokines, may have a vital role in insulin resistance and inflammation via inhibition of tyrosine kinase enzyme of insulin receptors in hepatic cells [70].

Histopathological examination of the kidney tissue in this study showed that STZ caused tubular injury in the kidney cortex, expanded vacuolar degeneration of renal epithelium of renal tubules, and expanded mesangial matrix. Similar inflammatory changes were also seen in pancreatic and spleen tissues after STZ injection in the study. However, noticeable improvements in histopathological changes in the liver, kidneys, pancreas, and spleen were seen in the animal group that received a combination of NRX + SITA in comparison with the STZ diabetic group. There were normally radiated hepatic cords, normal renal tissues, normal pancreas, and normal spleen tissue without characteristic lesions of diabetes.

The immunohistochemical study demonstrated that the liver tissues of the diabetic animals had marked expression of NF-κB, which was in agreement with Alqahtani et al. [71], who indicated that STZ single injection strongly increased the expression of NF-κB. The combined treatment of NRX + SITA in rats showed slight expression of hepatic NF-κB, which is in agreement with Alqahtani et al. who indicated that sitagliptin 100 mg/kg orally showed a decrease in NF-κB expression and reported that sitagliptin had a hepatoprotective effect [71].

NF-κB has an important role in inflammation and innate immunity. It also participates in the pathogenesis of diabetes and its complications. In diabetic patients, NF-κB is activated by IL-18, IL-6, and IL-1β. In diabetes type 1, pancreatic β-cell apoptosis is induced by NF-κB activation that is mediated via IL-1β [72]. However, activation of NF-κB in diabetes type 1 causes both insulin resistance and apoptosis. Advanced glycation end products (AGEs) and reactive oxygen species have an important role in the progression of diabetic complications. The upregulation of NF-κB may occur following the binding of AGEs and their receptors. Continuous NF-κB activation may induce systemic inflammation that forms an important risk factor for the occurrence of complications [73]. NF-κB was detected in atherosclerosis lesions in endothelial cells, macrophages, and vascular smooth muscles. NF-κB activation increases gene expression for many adhesion molecules, such as vascular cell adhesion molecule-l, which induces monocyte recruitment in the subendothelial space and constitutes an early step in the development of atherosclerosis [74].

## 5. Limitations of the Study

The primary limitation of this study is the lack of comprehensive evaluation of potential off-target effects and toxicity, which are critical for ensuring the safety and efficacy of the proposed approach. Addressing this limitation requires systematic toxicity assessments, including in vitro and in vivo studies to evaluate cytotoxicity, hemocompatibility, and immunogenicity. Additionally, improving targeting strategies through ligand–receptor interactions, antibody conjugation, or stimuli-responsive drug release could enhance specificity and minimize unintended interactions. Further pharmacokinetic and biodistribution studies are necessary to understand drug metabolism, clearance, and tissue accumulation, thereby reducing long-term risks. Optimizing dosing through controlled-release formulations may help balance therapeutic efficacy while limiting systemic exposure. Finally, long-term safety monitoring, including assessments of chronic toxicity, inflammation, and nanoparticle accumulation, is essential to ensure the sustained safety and clinical applicability of this approach.

## 6. Conclusions

Nanotechnology-based drug delivery systems represent a promising frontier in the management of complex diseases like diabetes. In this study, nano-rivaroxaban demonstrated significant immunomodulatory effects by reducing pro-inflammatory and anti-inflammatory cytokine levels, improving histopathological alterations in key organs, and suppressing hepatic NF-κB expression in STZ-induced diabetic rats. Notably, the combination of NRX and SITA exhibited synergistic effects, offering superior restoration of immune balance, enhanced organ protection, and better control of diabetes-associated inflammation compared to monotherapy.

These findings suggest that integrating nanoparticle-based therapies with conventional treatments can not only improve glycemic control but also address the underlying inflammatory processes driving diabetes complications, thereby reducing the risk of thrombosis and organ damage. Further studies are warranted to explore the translational applicability of these findings in clinical settings and to optimize the use of nanotechnology in diabetes care.

## Figures and Tables

**Figure 1 diseases-13-00087-f001:**
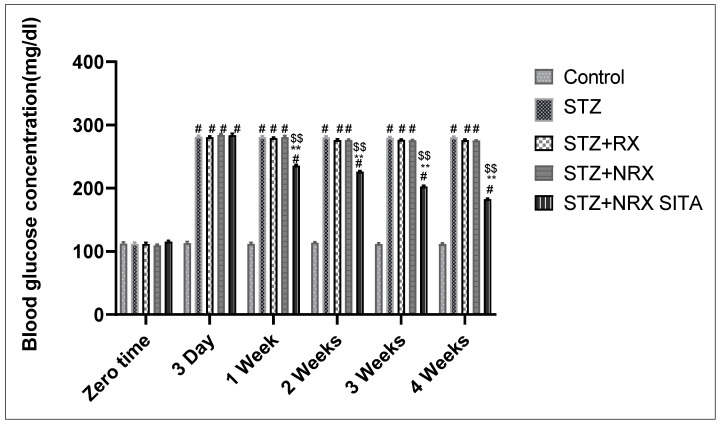
Effect of streptozotocin (STZ) (60 mg/kg single i.p.), rivaroxaban (RX) (5 mg/kg orally for 4 weeks), nano-rivaroxaban (NRX), and combined treatment of NRX + SITA (10 mg/kg orally for 4 weeks) on blood glucose level in diabetic rats. Data represented as mean ± SE of ten observations. ^#^ Significant difference at *p* < 0.001 in contrast to the control group. ** Significant difference at *p* < 0.001 in contrast to STZ group. ^$$^ Significant difference at *p* < 0.001 in contrast to NRX group.

**Figure 2 diseases-13-00087-f002:**
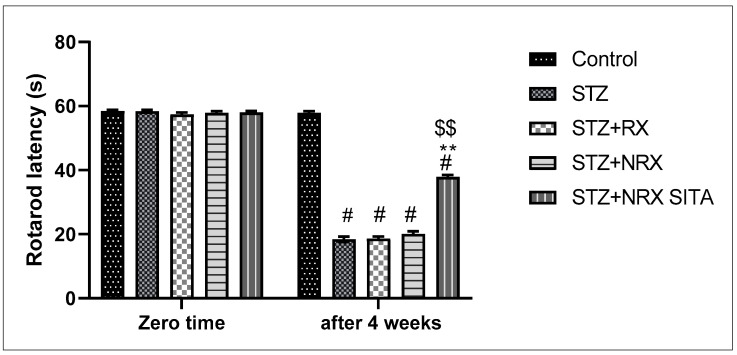
Effect of streptozotocin (STZ) (60 mg/kg single i.p.), rivaroxaban (RX) (5 mg/kg orally for 4 weeks), nano-rivaroxaban (NRX), and combined treatment of nano-rivaroxaban and sitagliptin (10 mg/kg orally for 4 weeks) (NRX + SITA) on rotarod staying time. Data represented as mean ± SE of ten observations. ^#^ Significant difference at *p* < 0.001 in contrast to the control group. ** Significant difference at *p* < 0.001 in contrast to STZ group. ^$$^ Significant difference at *p* < 0.001 in contrast to NRX group.

**Figure 3 diseases-13-00087-f003:**
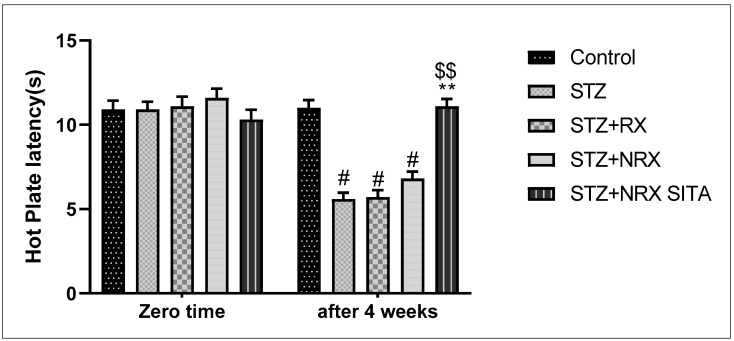
Effect of streptozotocin (STZ) (60 mg/kg single i.p.), rivaroxaban (RX) (5 mg/kg orally for 4 weeks), nano-rivaroxaban (NRX), and combined treatment of nano-rivaroxaban and sitagliptin (10 mg/kg orally for 4 weeks) (NRX + SITA) on hot plate latency. Data represented as mean ± SE of 10 observations. ^#^ Significant difference at *p* < 0.001 in contrast to the control group. ** Significant difference at *p* < 0.001 in contrast to STZ group. ^$$^ Significant difference at *p* < 0.001 in contrast to NRX group.

**Figure 4 diseases-13-00087-f004:**
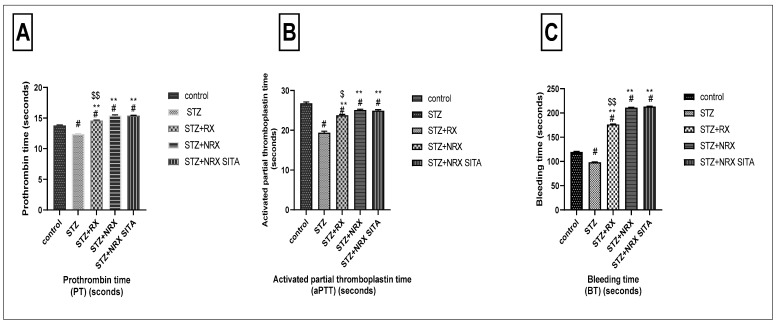
Effects of streptozotocin (STZ) (60 mg/kg single i.p.), rivaroxaban (RX) (5 mg/kg orally for 4 weeks), nano-rivaroxaban (NRX), and combined treatment of nano-rivaroxaban and sitagliptin (10 mg/kg orally for 4 weeks) (NRX + SITA) on coagulation profile. (**A**) Prothrombin time, (**B**) Activated partial thromboplastin time, (**C**) Bleeding time. Data represented as mean ± SE of 10 observations. ^#^ Significant difference at *p* < 0.001 in contrast to the control group. ** Significant difference at *p* < 0.001 in contrast to the STZ group. ^$^ Significant difference at *p* < 0.05 in contrast to NRX group. ^$$^ Significant difference at *p* < 0.001 in contrast to NRX group.

**Figure 5 diseases-13-00087-f005:**
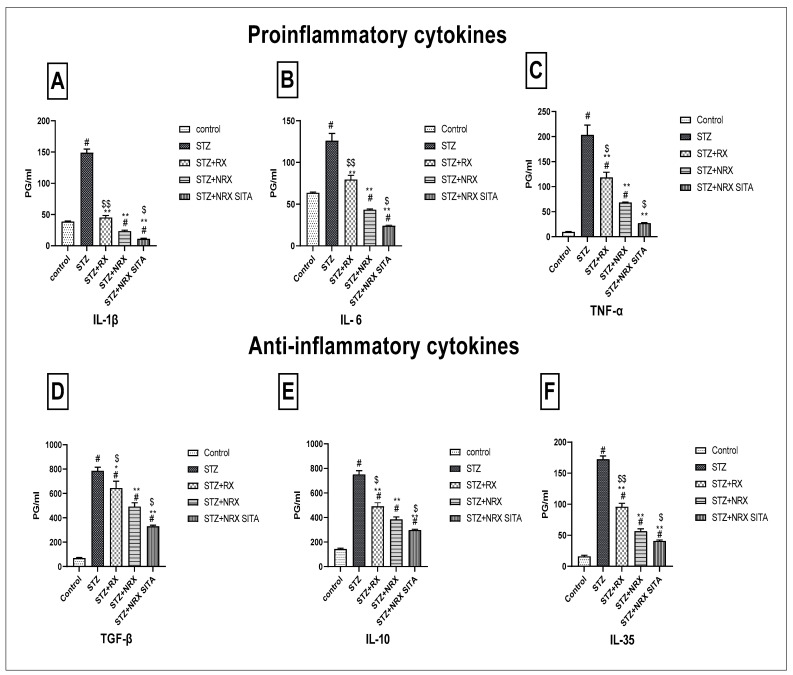
Effects of streptozotocin (STZ) (60 mg/kg single i.p.), rivaroxaban (RX) (5 mg/kg orally for 4 weeks), nano-rivaroxaban (NRX), and combined treatment of nano-rivaroxaban and sitagliptin (10 mg/kg orally for 4 weeks) (NRX + SITA) on cytokines (**A**) IL-1, (**B**) IL-6, (**C**) TNF-α, (**D**) TGF-β, (**E**) IL10, and (**F**) IL-35 in rat serum. Data represented as mean ± SE of ten observations. ^#^ Significant difference at *p* < 0.001 compared with the control group. * Significant difference at *p* < 0.05 compared with the STZ group. ** Significant difference at *p* < 0.001 compared with the STZ group. ^$^ Significant difference at *p* < 0.05 compared with the NRX group. ^$$^ Significant difference at *p* < 0.001 compared with the NRX group.

**Figure 6 diseases-13-00087-f006:**
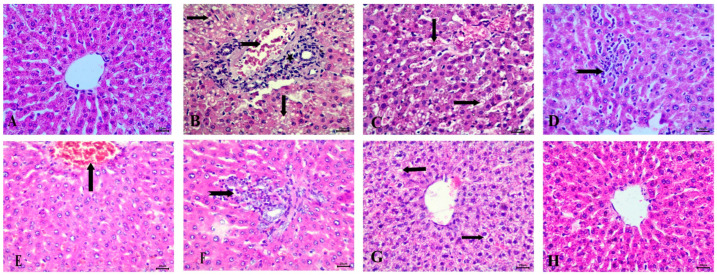
Histopathological examination of the liver stained with Hematoxylin and Eosin (H&E). (**A**) control group with normal architecture, (**B**) STZ-induced diabetic animals with fat vacuoles in hepatocytes (arrows), congestion of blood vessels (notched arrow), and peri-portal infiltration with inflammatory cells. (**C**) Diabetic animals with congestion of hepatic sinusoids (arrows). (**D**) STZ-induced diabetic group showing Kupffur cell proliferation (notched arrow). (**E**) RX-treated group showing congestion of central vein, (**F**) RX-treated group showing peri-portal lymphocytic infiltration (notched arrow), (**G**) NRX-treated group showing vacuolar degeneration (arrows), and (**H**) NRX + SITA-treated group showing normal hepatic cells. Scale bar = 20 µm.

**Figure 7 diseases-13-00087-f007:**
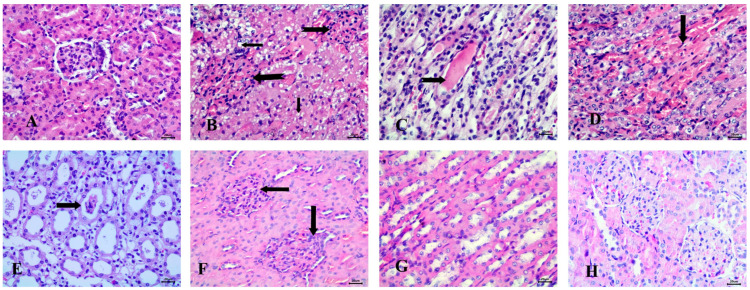
Histopathological examination of the kidney stained with Hematoxylin and Eosin (H&E). (**A**) control animals with normal glomerulus and renal tubules, (**B**) STZ-induced diabetic animals with fat vacuoles in tubular epithelium (notched arrows), absence of glomerular space (notched arrows), (**C**) Diabetic animals with hyaline casts (notched arrow), (**D**) Diabetic animals with interstitial hyperemia (arrow), (**E**) RX-treated group showing epithelial casts (arrow), (**F**) NRX-treated rats with absence of glomerular space (arrows), (**G**) STZ + NRX-treated animals with normal medullary tubules, and (**H**) NRX + SITA-treated group showing normal glomerular structure. Scale bar = 20 µm.

**Figure 8 diseases-13-00087-f008:**
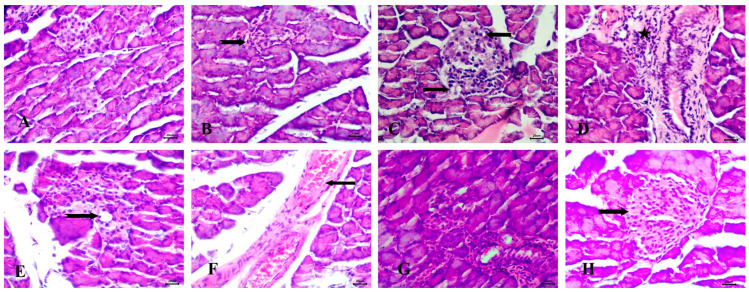
Histopathological examination of the pancreas stained with Hematoxylin and Eosin (H&E). (**A**) control group with normal Langerhans islets with pale rounded and ovoid β-cells in the center, (**B**) Diabetic animals with atrophy of Langerhans islets (notched arrow), (**C**) Diabetic rats with vacuolar degeneration of Langerhans islets cells (arrows), (**D**) Diabetic animals with inflammatory cellular infiltration peri-lobular duct (star), (**E**) RX-treated rats with mild vacuolar degeneration in cells of Langerhans islets (notched arrow), (**F**) RX-treated animals with congestion of interstitial blood vessels (arrow), (**G**) NRX-treated rats with atrophy of Langerhans islets (notched arrow), and (**H**) NRX + SITA-treated animals with normal islets of Langerhans (arrow). Scale bar = 20 µm.

**Figure 9 diseases-13-00087-f009:**
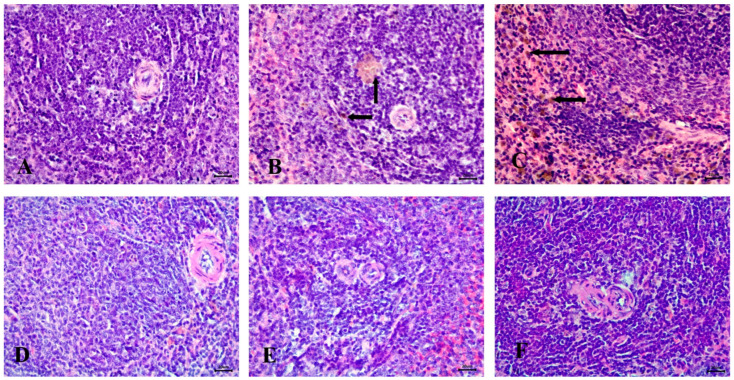
Histopathological examination of the spleen stained with Hematoxylin and Eosin (H&E). (**A**) control animals with healthy white pulp, (**B**) STZ-induced diabetic animals with severe hemosiderosis in red pulp (arrow), (**C**) STZ-induced diabetic animals with severe hemosiderosis in white pulp (arrow), (**D**) RX-treated group, (**E**) NRX-treated group, and (**F**) NRX + SITA-treated group showing normal white pulp. Scale bar = 20 µm.

**Figure 10 diseases-13-00087-f010:**
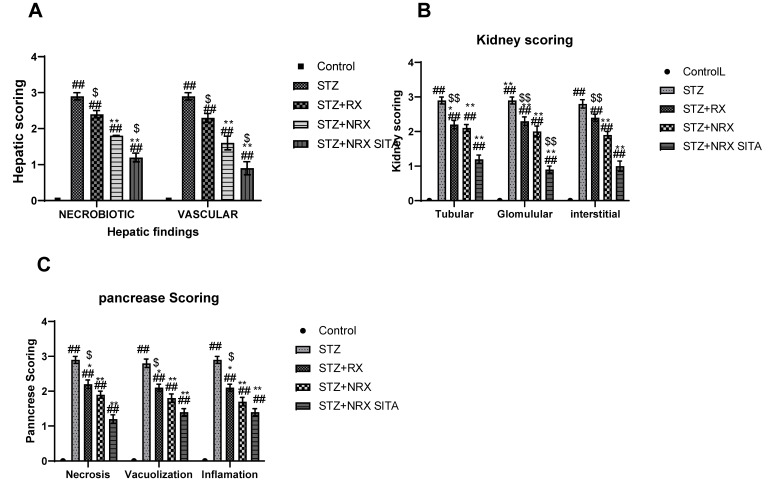
Effects of streptozotocin (STZ) (60 mg/kg single i.p.), rivaroxaban (RX) (5 mg/kg/day orally for 4 weeks), nano-rivaroxaban (NRX), and combined treatment of nano-rivaroxaban and sitagliptin (10 mg/kg/day orally for 4 weeks) (NRX + SITA) on histopathological scoring: (**A**) liver, (**B**) kidney, (**C**) pancreas in rats. Data represented as mean ± SE of ten observations. ^##^ Significant difference at *p* < 0.001 in contrast to the control group. * Significant difference at *p* < 0.05 in contrast to the STZ group. ** Significant difference at *p* < 0.001 in contrast to STZ group. ^$^ Significant difference at *p* < 0.05 in contrast to NRX group. ^$$^ Significant difference at *p* < 0.001 in contrast to NRX group.

**Figure 11 diseases-13-00087-f011:**
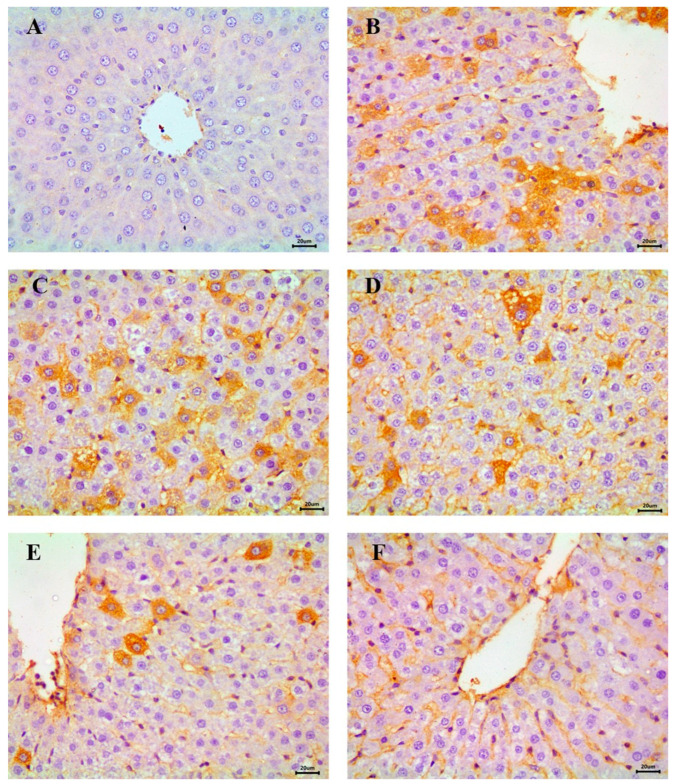
Immunohistochemical examination of NF-κB in the liver showing (**A**) control animals with negative expression, (**B**,**C**) STZ-induced diabetic group with a severe expression, (**D**) RX-treated group with moderate expression, (**E**) NRX-treated group with moderate expression, and (**F**) NRX + SITA-treated group with a mild expression, scale bar = 20 µm.

**Table 1 diseases-13-00087-t001:** Characterization of RX/PLGA NPs at different weight ratios.

RX/PLGA (wt Ratio)	Particle Size (nm)	PDI	Entrapment Efficiency (%)
1:1	995 ± 299	0.83 ± 0.16	96 ± 0.3
1:3	379 ± 15.0	0.43 ± 0.04	97 ± 0.3
1:5	1219 ± 182	1.00 ± 0.00	97 ± 0.4

## Data Availability

Data are contained within the article.

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
