# Peer review of "Immunomodulatory Effect of Rivaroxaban Nanoparticles Alone and in Combination with Sitagliptin on Diabetic Rat Model"

_diseases, 2025, doi:10.3390/diseases13030087_

Round 1

Reviewer 1 Report

Comments and Suggestions for Authors
  1. How the dose of RX and NRX is optimized??
  2. What was the purpose of making RX into NRX?? If the nano form is more efficient, do we really need same amount of dose  or reduced dose??
  3. Since nanosized drugs could have some off target effects including more toxicity, how the authors are going to account these possible unwanted effects?
  4. It would have been more clear if the authors can  include some toxicity markers in the circulation such as ALP, ALP, ALT, LDH, serum creatinine etc
  5. what was the inclusion/exclusion criteria of the animals in the experimental group. This should be given in the manuscript.
  6. Discussion should be written more precisely based on experimental results.
  7. Fig. 2 should indicate that the assay was done with serum.
  8. Some words are redundant in the manuscript.

Author Response

  1. How the dose of RX and NRX is optimized??

Response: According to Daci et al. (2020), the dose of rivaroxaban was administered and included in the manuscript. The dose of nano rivaroxaban was determined through a clinical trial.

  1. What was the purpose of making RX into NRX?? If the nano form is more efficient, do we really need same amount of dose or reduced dose??

Response:

Nanorivaroxaban enabled targeted drug delivery to specific cells or tissues in the body, improved pharmacokinetics, enhanced drug efficacy, and minimized adverse effects by reducing the frequency of administration. In this study, treatment with nano rivaroxaban at the same dose demonstrated significant immunomodulatory effects. It reduced levels of both pro-inflammatory and anti-inflammatory cytokines, improved histopathological changes in key organs, and suppressed hepatic NF-κB expression in STZ-induced diabetic rats. Thus, the administration of the same dose proved to have more beneficial effects that can enable to reduce the dose thus reducing the side effects.

  1. Since nanosized drugs could have some off target effects including more toxicity, how the authors are going to account these possible unwanted effects?

Response: Thanks for the note. We totally agree with the reviewer but due to limited fund we couldn’t account the off-target effects of the nanosized drugs. So we add the following paragraph to the limitation of the study.

Limitation of the study:

This study would be better if could address potential off-target effects and toxicity through:

1-Toxicity Studies by conducting in vitro and in vivo assessments, including cytotoxicity, hemocompatibility, and immunogenicity tests.

2-Targeting strategies using ligand-receptor interactions, antibody conjugation, or stimuli-responsive drug release to enhance specificity.

3-Pharmacokinetics and biodistribution by evaluating drug metabolism, clearance, and tissue accumulation to minimize long-term risks.

4-Optimized dosing by designing controlled-release formulations to balance efficacy and reduce systemic exposure.

5- Long-term safety monitoring by assessing chronic toxicity, inflammation, and nanoparticle accumulation over time.

  1. It would have been more clear if the authors can include some toxicity markers in the circulation such as ALP, ALP, ALT, LDH, serum creatinine etc

Response: It was better to measure liver and kidney function tests and CBC to detect any toxicity for the nanoparticles on theses organs but unfortunately no blood samples available now to measure these parameters in this study due to plenty of tests done in this study. In any future study measuring of these parameters will be essential. 

  1. what was the inclusion/exclusion criteria of the animals in the experimental group. This should be given in the manuscript.

Response: Inclusion and exclusion criteria for the animals added to the manuscript page 3 line 114-118

  1. Discussion should be written more precisely based on experimental results.

Response: Thanks for the note. we revised the manuscript and did our best.

  1. Fig. 2 should indicate that the assay was done with serum.

Response: we added serum in the title of figure 2

  1. Some words are redundant in the manuscript.

Response: Thanks for the note. we revised the manuscript and did our best.

Reviewer 2 Report

Comments and Suggestions for Authors

Mohamed M. Elbadr et al have presented the study very effectively. This work can be accepted in this journal.

Before considering this manuscript in the Disease journal authors need to address minor concerns

  1. Enlist the similar reported combinations in the introduction section
  2. What was the volume of acetone
  3. Please mention drug loading in the revised manuscript
  4. Did the author measure the zeta potential of nano formulation? Please mention in the revised manuscript
  5. What are the limitations of the present study

Author Response

Reviewer 2:

  1. Elbadr et al have presented the study very effectively. This work can be accepted in this journal.

Before considering this manuscript in the Disease journal authors need to address minor concerns

 Enlist the similar reported combinations in the introduction section

Response: Thanks for the reviewer. we introduced a new paragraph in the manuscript in lines 66 to 72, page 2.

  1. What was the volume of acetone

Response: The volume of acetone was added in the manuscript.

  1. Please mention drug loading in the revised manuscript

Response: Thanks for this comment, and I agree that drug loading is an important parameter in the characterization of NPs. But we thought that drug entrapment is more important in this study, and in our further studies we plan to expand the experimental studies to cover all the aspects including drug loading.

  1. Did the author measure the zeta potential of nanoformulation? Please mention in the revised manuscript

Response: As we mentioned in the previous comment, in our further studies we plan to expand the experimental studies including zeta potential.

  1. What are the limitations of the present study

Response: we added the following paragraph to the revised manuscript.

limitations of the study:

This study would be better if could address potential off-target effects and toxicity through:

1-Toxicity studies by conducting in vitro and in vivo assessments, including cytotoxicity, hemocompatibility, and immunogenicity tests.

2-Targeting strategies using ligand-receptor interactions, antibody conjugation, or stimuli-responsive drug release to enhance specificity.

3-Pharmacokinetics and biodistribution by evaluating drug metabolism, clearance, and tissue accumulation to minimize long-term risks.

4-Optimized dosing by designing controlled-release formulations to balance efficacy and reduce systemic exposure.

5- Long-term safety monitoring by assessing chronic toxicity, inflammation, and nanoparticle accumulation over time.

Round 2

Reviewer 1 Report

Comments and Suggestions for Authors

Limitations of the study should be rewritten as a single paragraph rather than bulleted points. 

Comments on the Quality of English Language

English language should be edited for a better clarity

Author Response

Comments and Suggestions for Authors

Limitations of the study should be rewritten as a single paragraph rather than bulleted points.

Response: We appreciate the reviewer’s suggestion to present the limitations of the study in a more cohesive format. In response, we have revised the section to a single, well-structured paragraph that improves readability and ensures a smooth flow of ideas.

Comments on the Quality of English Language

English language should be edited for a better clarity

Response: We sincerely appreciate the reviewer’s valuable feedback regarding the clarity of the English language throughout the manuscript. In response, we have carefully revised the entire manuscript to enhance the clarity, coherence, and readability of the text.